# mULLER: A Modular Monad-Based Semantics of the Neurosymbolic ULLER Framework

**Daniel Romero Schellhorn**                     D.SCHELLHORN@UNI-OSNABRUECK.DE

**Till Mossakowski**                     TILL.MOSSAKOWSKI@UNI-OSNABRUECK.DE

University of Osnabrück, Osnabrück, Germany

**Editors:** Leilani H. Gilpin, Eleonora Giunchiglia, Pascal Hitzler, and Emile van Krieken

## Abstract

**ULLER** (Unified Language for LEarning and Reasoning) offers a unified first-order logic (FOL) syntax, enabling its knowledge bases to be used directly across a wide range of neurosymbolic systems. The original specification endows this syntax with three pairwise independent semantics—classical, fuzzy, and probabilistic—each accompanied by dedicated semantic rules. We show that these seemingly disparate semantics are all instances of one categorical framework based on *monads*, the very construct that models side effects in functional programming. This enables the *modular* addition of new semantics and systematic translations between them. As example, we outline the addition of generalized quantification in Logic Tensor Networks (LTN) to arbitrary (also infinite) domains by extending the Giry monad to probability spaces. In particular, our approach allows a modular implementation of ULLER in Python and Haskell, of which we have published initial versions on GitHub.

## 1. Introduction

Neurosymbolic integration is a rapidly developing branch of AI. In the past, numerous heterogeneous approaches have emerged, each with its own code base. [15] introduces ULLER, a unified neurosymbolic library that aspires to play for neurosymbolic systems the role that TensorFlow and PyTorch play for deep-learning workflows. Their theoretical core is the concept of a NeSy system: standard first-order logic enriched with neural components. In particular, formulas of the form

$$x := m(T_1, \ldots, T_n)(F) \qquad (T_i \text{ terms, } F \text{ a formula involving } x, m \text{ a neural model})$$

are used to integrate neural models $m$ into logical formulas. These formulas go beyond classical first-order logic. Instead, they perform computations that may return multiple values and typically involve non-determinism or probability distributions as illustrated in the following toy example [15]:

$$\forall x \in \textit{ImageData}$$
$$(n_1 := \textit{classify}(x.im_1)$$
$$(n_2 := \textit{classify}(x.im_2)$$
$$(n_1 + n_2 = x.sum) \,))$$

This specifies to classify two images of digits and to check whether the sum of the resulting numbers is as specified in the dataset. The resulting (e.g. fuzzy or probabilistic) truth value can be used in a loss function. A shorthand notation for this is:

$$\forall x \in \mathit{ImageData}\big(\; n_1 := \mathit{classify}(x.im_1), n_2 := \mathit{classify}(x.im_2)\; (n_1 + n_2 = x.sum)\;\big)$$

While the notion of NeSy system in [15] is very powerful, it also has several shortcomings:

- There is no uniform inductive definition of truth, i.e. of the truth value of a sentence in an interpretation. Rather, the notion of NeSy system has the inductive interpretation function as a component, meaning that classical, probabilistic and fuzzy NeSy systems employ three different inductive definitions of truth. Parts of these definitions of truth are copied verbatim from one NeSy system to another, other parts need to be replaced. This duplication of semantic rules is not modular. By contrast, we aim at a truly uniform inductive definition of truth value that is independent of the NeSy system and hence can be reused for different NeSy systems, such that the NeSy system itself is a parameter of the inductive definition of truth.

- The case of continuous probability distributions (involving probability kernels or Markov kernels) is not covered faithfully, because in this case, measurable spaces are required to properly define the mentioned Markov kernels. However, measurable spaces are not considered in [15].

- The treatment of logical connectives is not uniform across NeSy systems, i.e. the different sets of connectives are not considered as instances of a common abstract (algebraic) notion. Also, quantifiers in probabilistic semantics are defined using possibly infinite products, without requiring a suitable order structure on domains and without discussing convergence.

- The high-level concepts of semantics and computation are not properly separated. Computation (namely sampling) is mixed into the semantics at least in two places. The first one in the classical semantics, where the possibly multi-valued $\arg\max$ can only be properly evaluated using sampling. We conceptualize the $\arg\max$ differently as a transition *between* semantics. The second time is in "Sampling Semantics", which is not really semantics but computation (sampling).

We argue that ULLER is conceptually robust and show that a monadic formulation resolves all of the foregoing issues. In particular, ULLER formulas of the form

$$x := m(T_1, \ldots, T_n)(F) \qquad (T_i \text{ terms, } F \text{ a formula involving } x, m \text{ a neural model})$$

can be modeled using Moggi's notion of computational monad [9], which has been introduced to model side effects in (functional) programming. Although monads originate in category theory, we present them using a set-theoretic approach that does not involve any category theory. The generalization to an arbitrary category, which is needed for continuous probabilities and some aspects of infinite domains, has been relegated to the appendix. We call our categorical approach "monadic ULLER", "modular ULLER", or simply "$mULLER$"[1].

---

1. *muller*: noun, a heavy tool of stone or iron used to grind and mix material.

## 2. Set-Based NeSy Frameworks

A neurosymbolic framework (NeSy framework) is a general framework for NeSy systems combining neural models with symbolic logic, and it provides the semantic background for the specific logic involved. Examples are the logics behind DeepProbLog [8] or Logic Tensor Networks [3].

The notion of NeSy framework is not defined in [15]. Rather, they define a notion of NeSy system, which is quite ad-hoc, because it simultaneously makes two choices: (1) a choice of a particular interpretation with functions, predicates and neural models (e.g., probabilities for traffic lights, or neural networks learning addition of digit images), and (2) a choice of semantic rules for interpreting terms (which also involves a choice of the logic, e.g. classical or probabilistic or fuzzy). This causes semantic rule duplication.

Our notion of NeSy framework provides a means to disentangle these two choices. Moreover, our approach makes semantic rules independent not only of particular interpretations, but also of the choice of logic (classical, probabilistic, or fuzzy).

In the sequel, we first introduce some background on monads, which are the key concept of our approach, and on the algebraic structure needed to model the space of truth values. Then we go on to define the notions of NeSy framework and NeSy system.

### 2.1. Set-Based Monads

We interpret formulas $x := m(T_1, \ldots, T_n)(F)$ involving neural models $m$ (= certain computations) using Moggi's notion of computational monad [9] in the form of Kleisli triples:

**Definition 1** *A **Kleisli triple** (monad) $(\mathcal{T}, \eta, (-)^*)$ consists of:*

- *A mapping $\mathcal{T}$, mapping sets $X$ to sets $\mathcal{T}X$ (of computations with values from $X$),*

- *A family of functions: $\eta_X : X \to \mathcal{T}X$ for each set $X$ (construing a value $a \in X$ as stateless computation $\eta_X(a) \in \mathcal{T}X$),*

- *A function that assigns to each function $f : X \to \mathcal{T}Y$ a function $f^* : \mathcal{T}X \to \mathcal{T}Y$ (called the* Kleisli extension*), needed for sequential composition of computations,*

*such that the following axioms hold:*

1. *$(\eta_X)^* = \mathrm{id}_{\mathcal{T}X}$,*

2. *$f^* \circ \eta_X = f$ for all $f : X \to \mathcal{T}Y$,*

3. *$(g^* \circ f)^* = g^* \circ f^*$ for all $f : X \to \mathcal{T}Y$ and $g : Y \to \mathcal{T}Z$.*

Given computations $ma : \mathcal{T}A$ and $mb(x) : \mathcal{T}B$, we can compose them to $(\lambda x : A.mb(x))^*(ma)$ of type $\mathcal{T}B$. In Haskell's do-notation, this is written as **do** $x \leftarrow ma$; $mb(x)$.

**Example 1** ***Non-empty Powerset monad** $\mathcal{P}_{\neq \emptyset}$ For a set $X$:*

$$\mathcal{P}_{\neq \emptyset}X := \{A \subseteq X \mid A \neq \emptyset\} \quad \text{(non-empty subsets of } X\text{)},$$

$$\eta_X(x) := \{x\}, \quad f^*(A) := \bigcup_{a \in A} f(a) \quad \big(f : X \to \mathcal{P}_{\neq \emptyset}Y, A \in \mathcal{P}_{\neq \emptyset}X\big).$$

**Example 2** *Probability distribution monad (Kleisli triple) $\mathcal{D}$.*[2]

$$\mathcal{D}X := \Big\{ \rho : X \to [0,1] \text{ finitely supported } \Big| \sum_{x \in X} \rho(x) = 1 \Big\} \quad \text{(prob. distributions on } X\text{)},$$

$$\eta_X(x) := \delta_x, \ \delta_x(y) = \begin{cases} 1, & x = y \\ 0, & x \neq y \end{cases} \quad f^*(\rho)(y) := \sum_{x \in X} f(x)(y) \cdot \rho(x) \quad \big( f : X \to \mathcal{D}Y, \ \rho \in \mathcal{D}X \big).$$

*$\delta_x$ is the probability distribution that assigns all probability mass to $x$. $f^*(\rho)$ corresponds to a two-level random process: first $x$ is drawn from $\rho$, then $y$ is drawn from $f(x)$. This results in a marginal distribution of $Y$ for the joint distribution $\hat{\rho}(x,y) := f(x)(y) \cdot \rho(x)$.*
***do*** *$x \leftarrow ma$; $mb(x)$ can be interpreted as "sample $x$ from $ma$ and then proceed with $mb(x)$".*

## 2.2. Double Semigroup Bounded Lattices (2Sg-Bl)

We need an algebraic structure to model the space of truth values. We weaken the notion of BL algebra of [6] from fuzzy logic as follows:

**Definition 2** *A **double semigroup bounded lattice (2Sg-Bl)** $\mathcal{R}$ is a tuple*

$$\big( S, \ \leq, \ \bot, \ \top, \ \otimes, \ \oplus, \ \to, \ \neg \big)$$

*in which $S$ is a set, $\mathcal{L} := (S, \leq)$ a bounded lattice, while $\bot \in S$ and $\top \in S$ are its bottom and top elements.[3] Also $(S, \otimes)$ and $(S, \oplus)$ are semigroups, $\to$ is a map $S \times S \to S$, and $\neg$ is a map $S \to S$.*

Also, we want to allow different aggregation operations other than infinite meet and join to cover the quantifiers of Logic Tensor Networks [3], motivating the following definition:[4]

**Definition 3** *An **aggregated 2Sg-Bl** has for each set $X$ two order-preserving maps:*

$$\text{aggr}_X^\forall, \text{aggr}_X^\exists : \ \mathcal{L}^X \longrightarrow \mathcal{L}.$$

*In case of a complete lattice, $\text{aggr}_X^\forall$ can be chosen as meet $\bigwedge_X$ and $\text{aggr}_X^\exists$ as join $\bigvee_X$.*

## 2.3. Definition of Set-Based NeSy Framework

Given some basic notion of truth $\Omega$, our NeSy systems work on the monadic space of truth values $\mathcal{T}\Omega$, which is required to be an aggregated 2Sg-Bl. If $\mathcal{T}$ is the identity monad, $\mathcal{T}\{0,1\}$ is just the two-element set $\{0,1\}$ of classical truth values. If $\mathcal{T}$ is the distribution monad, $\mathcal{T}\{0,1\}$ is isomorphic to the unit interval $[0,1]$, regarded as the space of probabilistic or fuzzy truth values.

---

2. Note that the sums below are only finite if one excludes all the zero addenda.

3. In many cases we have $\bot$ is neutral element for $\oplus$ and $\top$ is neutral element for $\otimes$, for example inside of the unit interval $[0,1]$. In some cases, like Gödel logic, we even have $\oplus = \vee$ and $\otimes = \wedge$. Also, $\to$ is normally chosen as right adjoint to $\otimes$ or as $x \to y := \neg x \oplus y$. In the first case $\neg$ can be defined as implication to zero, in the second one it is defined independently. Check table B.2 for details.

4. This is inspired by the notion of *aggregated functions* in [4].

**Definition 4** *A **NeSy framework** $\mathcal{F} = (\mathcal{T}, \Omega, \mathcal{R})$ consists of*

1. *a* monad $\mathcal{T}$,

2. *A* set $\Omega$ *acting as truth basis,*[5]

3. *an aggregated* 2Sg-Bl $\mathcal{R}$ *on the truth space* $\mathcal{T}\Omega$.

Examples are given in Table 1 and discussed in more detail in section 4. Note that further examples arise by varying the 2Sg-Bl $\mathcal{R}$ on $[0, 1]$. Examples requiring category theory are given in Table 4 in the appendix.

Table 1: NeSy Framework Examples (set-based)

| **Logic/Theory** | $\mathcal{T}$ | $\Omega$ | $\mathcal{T}\Omega$ | $\mathcal{R}$ | **Subsection** |
|---|---|---|---|---|---|
| Classical | Identity | $\{0, 1\}$ | $\{0, 1\}$ | Boolean Alg. | 4.1 |
| Three-valued LP | Powerset $\mathcal{P}_{\neq \emptyset}$ | $\{0, 1\}$ | $\{0, B, 1\}$ | Kleene/Priest Alg. | 4.1 |
| Distributional | Distribution $\mathcal{D}$ | $\{0, 1\}$ | $[0, 1]$ | Product BL–Alg. | 4.2 |
| Finitary $\mathrm{LTN}_p$ | Distribution $\mathcal{D}$ | $\{0, 1\}$ | $[0, 1]$ | Product SBL-Alg. | 4.3 |
| Classical Fuzzy | Identity | $[0, 1]$ | $[0, 1]$ | Classical BL–Alg. | – |

## 3. Syntax and Semantics of mULLER

### 3.1. Syntax of First-Order Logic

The ULLER language of [15] features computational function symbols that can be realized e.g. by neural networks. In a similar spirit, we here add computational predicate symbols, which are also realized by neural networks, for example in Logic Tensor Networks [3].

**Definition 5** *A **NeSy signature** $\Sigma$ consists of*

- *a set $S$ of* sorts *of $\Sigma$,*

- *two disjoint sets* Pred, MPred *of* predicate symbols *and* computational predicate symbols *of form $p : s_1, \ldots, s_n$, where $p$ is a name and each $s_i \in S$ a sort,*

- *two disjoint sets* Func, MFunc *of* function symbols *and* computational function symbols *of form $f : s_1, \ldots, s_n \to s$, , where $f$ is a name and $s, s_i \in S$ are sorts.*

*(Computational) predicate symbols with no arguments are called (computational) propositional symbols* (Prps *and* MPrps *respectively). Function symbols with one argument are called properties* (Prop)*, those with none are called constants* (Const)*.*

---

5. Similar to the basis of a vector space.

Concerning syntax, we largely follow the definitions given in [15]. That is, given a *signature* $\Sigma$ of non-logical symbols and set of variables V, we can define the syntax of first-order logic (FOL) formulas over $\Sigma$ and V as a context-free grammar:

**Terms:**

$$T ::= x : s \qquad\qquad\qquad\qquad\qquad [x \in \mathrm{V}, s \in S]$$

$$T ::= c \mid T.\mathsf{prop} \mid f(T, \ldots, T) \qquad\qquad [c \in \mathrm{Const}, \mathsf{prop} \in \mathrm{Prop}, f \in \mathrm{Func}]$$

**Atomic Formulas:**

$$F ::= R \mid P(T, \ldots, T) \qquad\qquad\qquad [R \in \mathrm{Prps}, P \in \mathrm{Pred}]$$

$$F ::= N \mid M(T, \ldots, T) \qquad\qquad\qquad [N \in \mathrm{MPrps}, M \in \mathrm{MPred}]$$

**Compound Formulas:**

$$F ::= \bot \mid \top \mid F \to F \mid \neg F \mid F\|F \mid F\&F \mid (F)$$

$$F ::= \exists x : s\ (F) \mid \forall x : s\ (F) \qquad\qquad\qquad [x \in \mathrm{V}, s \in S]$$

$$F ::= x := m(T, \ldots, T)(F) \qquad\qquad\qquad\quad [m \in \mathrm{MFunc}]$$

### 3.2. Tarskian Semantics

**Definition 6** *For a* NeSy *signature* $\Sigma$*, a **NeSy interpretation** $\mathcal{I}$ on a* set-based NeSy *framework* $(\mathcal{T}, \Omega, \mathcal{R})$ *is given by*

- *a set $\mathcal{I}(s)$ for every sort $s$,*

- *a function $\mathcal{I}(f) : \mathcal{I}(s_1) \times \ldots \times \mathcal{I}(s_n) \to \mathcal{I}(s)$ for every (normal) function symbol $f : s_1, \ldots, s_n \to s \in \mathrm{Func}$,*

- *a function $\mathcal{I}(m) : \mathcal{I}(s_1) \times \ldots \times \mathcal{I}(s_n) \to \mathcal{T}(\mathcal{I}(s))$ for every computational function symbol $m : s_1, \ldots, s_n \to s \in \mathrm{MFunc}$,*

- *a function $\mathcal{I}(P) : \mathcal{I}(s_1) \times \ldots \times \mathcal{I}(s_n) \to \Omega$ for every predicate symbol $P : s_1, \ldots, s_n \in \mathrm{Pred}$,*

- *and a function $\mathcal{I}(M) : \mathcal{I}(s_1) \times \ldots \times \mathcal{I}(s_n) \to \mathcal{T}\Omega$ for every computational predicate symbol $M : s_1, \ldots, s_n \in \mathrm{MPred}$.*

**Definition 7** *A **NeSy system** can be defined as a quadrubple $(\mathcal{T}, \Omega, \mathcal{R}, \mathcal{I})$, where $(\mathcal{T}, \Omega, \mathcal{R})$ is a* NeSy framework *and $\mathcal{I}$ a* NeSy interpretation *on that same* NeSy framework.

Compared to [15], we have added computational predicate symbols, because LTN and other NeSy frameworks use these. However note that for probabilistic logic and weighted model counting, ULLER makes independence assumptions due to the nature of its notion of interpretation.[6] While computational function symbols enable the use of conditional probabilities, computational predicate symbols are always independent of each other. Hence, ULLER supports a certain combination of probabilistic and fuzzy logic, and so do LTNs. For details, see Appendix C.1.

---

6. See [16] for a critical discussion.

Also, we have dropped uniformity of the notion of interpretation—it now becomes dependent on the monad at hand. This is necessary for faithfully distinguishing finitely supported and continuous probability distributions and for dealing with LTN-style quantification on infinite domains. Still, computational symbols can be realised by neural networks in all of these cases. However, the details of the mapping from neural networks to interpretations of computational symbols differ.

In [15], based on an interpretation, the notion of NeSy system provides a Tarskian inductive definition $\llbracket \cdot \rrbracket$ of the semantics of formulas and thus it implicitly also defines the semantics of the logical symbols. The drawback of this approach is that the Tarskian semantics $\llbracket \cdot \rrbracket$ is inherently tied to the specific NeSy system.

We can modularise matters here, because we first give a semantics of the logical symbols via a NeSy framework, and based on that, the interpretation provides the semantics of the non-logical symbols. Hence, the inductive definition of the Tarskian semantics $\llbracket \cdot \rrbracket$ needs to be given only once, and this definition holds across all NeSy frameworks and systems.

**Definition 8**  *The **Tarskian semantics** $\llbracket \cdot \rrbracket$ of a* NeSy *system* $(\mathcal{T}, \Omega, \mathcal{R}, \mathcal{I})$ *is given by:*

$$\textbf{Formulas: } \llbracket F \rrbracket_{\mathcal{I}} : \mathcal{V}_F \to \mathcal{T}\Omega, \quad \textbf{Terms: } \llbracket T \rrbracket_{\mathcal{I}} : \mathcal{V}_T \to \mathcal{I}(s_T).$$

Table 2: Inductive definition of the Tarskian semantics

| Syntax | Set Semantics $\llbracket \cdot \rrbracket_{\mathcal{I}, \nu}$ |
|---|---|
| **Terms** | |
| $\llbracket x : s \rrbracket$ | $\nu(x)$ |
| $\llbracket c \rrbracket$, $\llbracket T.\mathsf{prop} \rrbracket$, $\llbracket f(\boldsymbol{T}) \rrbracket$ | $\mathcal{I}(c)$, $\mathcal{I}(\mathsf{prop})(\llbracket \boldsymbol{T} \rrbracket)$, $\mathcal{I}(f)(\llbracket \boldsymbol{T} \rrbracket)$ |
| **Atomic formulas** | |
| $\llbracket P \rrbracket$, $\llbracket R(\boldsymbol{T}) \rrbracket$ | $\eta_\Omega(\mathcal{I}(P))$, $\eta_\Omega\big(\mathcal{I}(R)(\llbracket \boldsymbol{T} \rrbracket)\big)$ |
| $\llbracket N \rrbracket$, $\llbracket M(\boldsymbol{T}) \rrbracket$ | $\mathcal{I}(N)$, $\mathcal{I}(M)\big(\llbracket \boldsymbol{T} \rrbracket\big)$ |
| **Compound formulas** | |
| $\llbracket \bot \rrbracket$, $\llbracket \top \rrbracket$ | $\bot_{\mathcal{R}}$, $\top_{\mathcal{R}}$ |
| $\llbracket F \to G \rrbracket$, $\llbracket \neg F \rrbracket$ | $\llbracket F \rrbracket \to_{\mathcal{R}} \llbracket G \rrbracket$, $\neg_{\mathcal{R}} \llbracket F \rrbracket$ |
| $\llbracket F \| G \rrbracket$, $\llbracket F \& G \rrbracket$ | $\llbracket F \rrbracket \oplus_{\mathcal{R}} \llbracket G \rrbracket$, $\llbracket F \rrbracket \otimes_{\mathcal{R}} \llbracket G \rrbracket$ |
| $\llbracket \exists x{:}s\, F \rrbracket$, $\llbracket \forall x{:}s\, F \rrbracket$ | $\mathrm{aggr}^{\exists}_{\mathcal{I}(s)}(\lambda a.\llbracket F \rrbracket_{\nu[x \mapsto a]})$, $\mathrm{aggr}^{\forall}_{\mathcal{I}(s)}(\lambda a.\llbracket F \rrbracket_{\nu[x \mapsto a]})$ |
| $\llbracket x := m(\boldsymbol{T})(F) \rrbracket$ | $\textbf{do } a \leftarrow \mathcal{I}(m)(\llbracket \boldsymbol{T} \rrbracket); \llbracket F \rrbracket_{\nu[x \mapsto a]}$ |

*Here, we define $\mathcal{V}_T := \prod_{x:t \in \Gamma_T} \mathcal{I}(t)$, where $\Gamma_T$ is the context of $T$ and $s_T$ is the (unique) sort of the term $T$. Analogously $\mathcal{V}_F := \prod_{x:t \in \Gamma_F} \mathcal{I}(t)$. $\boldsymbol{T}$ stands for $T_1, \ldots, T_n$. We work with variable valuations $\nu \in \mathcal{V}_T$ (or $\nu \in \mathcal{V}_F$) in local (term or formula) contexts, noting that elements of $\prod_{x:t \in \Gamma_T} \mathcal{I}(t)$ map variables $x : t$ to values in $\mathcal{I}(t)$. We write $\llbracket T \rrbracket_{\mathcal{I}, \nu} = \llbracket T \rrbracket_{\mathcal{I}}(\nu)$ and $\llbracket F \rrbracket_{\mathcal{I}, \nu} = \llbracket F \rrbracket_{\mathcal{I}}(\nu)$. That said, we mostly omit $\mathcal{I}$ and $\nu$ if clear from the context.*

## 4. Examples of Set-Based Semantics

In the sequel, we will discuss some NeSy frameworks in more detail and spell out how the semantic rules look when instantiated. We often implicitly define parts of the 2Sg-Bl through the semantic rules. E.g. the $\text{aggr}^\exists$ and $\text{aggr}^\forall$ functions are implicitly defined by listing semantic rules for the quantifiers.

### 4.1. Classical and Three-valued Semantics

Classical semantics is simply given by the identity monad and the Boolean algebra on $\Omega = \{0,1\}$, which results in classical first-order logic.

Our classical semantics is deterministic, while [15] use probability distributions, causing the need for selection of values with highest probability, done via argmax. We will model this as NeSy transformation in section 5 and need a non-deterministic NeSy framework as target of this transformation. The (non-empty) powerset monad models non-deterministic computations, cf. multialgebras [18]. These result in non-deterministic truth values as in:

**Logic of Paradox Semantics** For the Logic of Paradox [11] with the non-empty powerset monad $\mathcal{P}_{\neq\emptyset}$, we have $\mathcal{T} = \mathcal{P}_{\neq\emptyset}$, $\Omega = \{0,1\}$ (equivalently $\{F,T\}$), and $\mathcal{T}\Omega = \mathcal{P}_{\neq\emptyset}(\{0,1\}) = \{\{0\},\{1\},\{0,1\}\}$. The three truth values correspond to: $\{0\} \equiv F$ (false only), $\{1\} \equiv T$ (true only), and $\{0,1\} \equiv B$ (both true and false). Following the uniform Tarskian semantics:

$$\llbracket x := m(\boldsymbol{T})(F) \rrbracket := \bigcup_{a \in \mathcal{I}(m)(\llbracket \boldsymbol{T} \rrbracket)} \llbracket F \rrbracket_{\nu[x \mapsto a]} \tag{1}$$

$$\llbracket \exists x{:}s\ F \rrbracket := \sup_{a \in \mathcal{I}(s)} \llbracket F \rrbracket_{\nu[x \mapsto a]}, \quad \llbracket \forall x{:}s\ F \rrbracket := \inf_{a \in \mathcal{I}(s)} \llbracket F \rrbracket_{\nu[x \mapsto a]} \tag{2}$$

$$\llbracket F \| G \rrbracket := \max(\llbracket F \rrbracket, \llbracket G \rrbracket), \quad \llbracket F \& G \rrbracket := \min(\llbracket F \rrbracket, \llbracket G \rrbracket) \tag{3}$$

$$\llbracket F \to G \rrbracket := \max(\llbracket \neg F \rrbracket, \llbracket G \rrbracket), \quad \llbracket \neg F \rrbracket := \begin{cases} \{0,1\} & \text{if } \llbracket F \rrbracket = \{0,1\} \\ (\{0,1\} \setminus \llbracket F \rrbracket) & \text{else} \end{cases} \tag{4}$$

$$\llbracket \bot \rrbracket := \{0\}, \quad \llbracket \top \rrbracket := \{1\} \tag{5}$$

where the operations implement Priest's Logic of Paradox with the lattice ordering $\{0\} <_{\text{LP}} \{0,1\} <_{\text{LP}} \{1\}$ (i.e., F < B < T).

### 4.2. Distributional Semantics

The distributional semantics corresponds to the third row in Table 1, where we use the distribution monad $\mathcal{D}$ over the classical truth basis $\{0,1\}$, yielding the truth space $[0,1]$ equipped with a Product BL-algebra structure. This framework provides the semantic foundation for probabilistic logic programming systems and neural-symbolic approaches that work with probability distributions over truth values. In this setting, computational predicates and function symbols return probability distributions rather than deterministic values. We need to restrict interpretations to finite domains. Finite quantification just iterates conjunction or disjunction. For infinite quantifiers, check appendix C.1.

$$\llbracket x := m(\boldsymbol{T})(F) \rrbracket := \sum_{a \in \mathcal{I}(s_m)} \llbracket F \rrbracket_{\nu[x \mapsto a]} \cdot \rho_m(a \mid \boldsymbol{T}) \tag{6}$$

$$\llbracket \exists x{:}s\ F \rrbracket := 1 - \prod_{a \in \mathcal{I}(s)} (1 - \llbracket F \rrbracket_{\nu[x \mapsto a]}), \quad \llbracket \forall x{:}s\ F \rrbracket := \prod_{a \in \mathcal{I}(s)} \llbracket F \rrbracket_{\nu[x \mapsto a]} \tag{7}$$

$$\llbracket F \| G \rrbracket := \llbracket F \rrbracket + \llbracket G \rrbracket - \llbracket F \rrbracket \cdot \llbracket G \rrbracket, \quad \llbracket F \& G \rrbracket := \llbracket F \rrbracket \cdot \llbracket G \rrbracket \tag{8}$$

$$\llbracket F \to G \rrbracket := \max\big(1, \llbracket G \rrbracket / \llbracket F \rrbracket\big), \quad \llbracket \neg F \rrbracket := 1 - \llbracket F \rrbracket \tag{9}$$

$$\llbracket \bot \rrbracket := 0, \quad \llbracket \top \rrbracket := 1. \tag{10}$$

### 4.3. Finitary LTN$_p$ Semantics

The finitary Logic Tensor Networks (LTN) semantics corresponds to the fourth row in Table 1, employing the distribution monad $\mathcal{D}$ over the classical truth basis $\{0, 1\}$ with the truth space $[0, 1]$ equipped with a Product Real Logic structure, what we call the S-Product Algebra in Table B.2. It is the same as the Product Algebra in the distributional semantics, only that we use S-Implication (strong implication) instead of R-Implication (residual implication). Moreover, aggregation differs as well: following [3], quantification is performed using $p$-norms, where the parameter $p$ controls the "softness" of the logical operations. For existential quantification, we compute the $p$-norm of truth values, while for universal quantification, we use the dual formulation $1 - \|1 - \cdot\|_p$:

$$\llbracket \exists x{:}s\ F \rrbracket := \Big( \frac{1}{|\mathcal{I}_s|} \sum_{a \in \mathcal{I}_s} \llbracket F \rrbracket^p_{\nu[x \mapsto a]} \Big)^{1/p}, \tag{11}$$

$$\llbracket \forall x{:}s\ F \rrbracket := 1 - \Big( \frac{1}{|\mathcal{I}_s|} \sum_{a \in \mathcal{I}_s} (1 - \llbracket F \rrbracket_{\nu[x \mapsto a]})^p \Big)^{1/p}, \tag{12}$$

$$\llbracket F \to G \rrbracket := 1 - \llbracket F \rrbracket + \llbracket F \rrbracket \cdot \llbracket G \rrbracket. \tag{13}$$

This is however only possible for finite domains. For the infinite case and additional quantifier variants, we refer to Appendix C.2.

### 4.4. Sampling "Semantics"

While [15] have introduced a sampling semantics, we think that the semantics should define probabilities, while an implementation can work with e.g. Monte Carlo sampling in order to obtain an approximation that is easier to implement (and, in the case of quantification over infinite domains, unavoidable). Hence, we do not discuss sampling semantics here. But we expect that a Monte Carlo convergence theorem can be stated and proved.

## 5. NeSy Transformations

The original ULLER paper [15] uses a uniform notion of interpretation. This leads to the problem that classical semantics needs to extract values with maximal probability (using

arg max) from a distribution, which is not possible if there is a tie[7]. Here, we propose an alternative way of dealing with this problem: namely, using a NeSy transformation, we can move e.g. from an interpretation in a probabilistic NeSy framework to one in a classical framework. Deadling with ties can be done using a non-deterministic semantics.

**Definition 9** *A **NeSy transformation** $\alpha : \mathcal{F} \to \mathcal{F}'$ between two NeSy frameworks is a family of functions*[8] *$\alpha_\Sigma : \mathrm{Intp}_\mathcal{F}(\Sigma) \to \mathrm{Intp}_{\mathcal{F}'}(\Sigma)$. Here, the* interpretation function $\mathrm{Intp}_\mathcal{F} : \Sigma \mapsto \mathrm{Intp}(\mathcal{F}, \Sigma)$ *sends a signature to the set of interpretations on $\mathcal{F}$ for that signature. We write $\alpha$ for $\alpha_\Sigma$ if $\Sigma$ is clear from context.*

**Argmax transformation: From distributional to non-deterministic semantics**
For a given distributional interpretation $\mathcal{I}(m)$ of a computational function symbol $m$, we can define a non-deterministic interpretation $\alpha(\mathcal{I})(m)$ of $m$ by defining, where $s_m$ is the sort of $m$, and likewise, $M$ is a computational predicate symbol:

$$\alpha(\mathcal{I})(m) := \underset{a \in \mathcal{I}(s_m)}{\arg\max}\big[\mathcal{I}(m)(a)\big], \qquad \alpha(\mathcal{I})(M) := \underset{b \in \Omega = \{0,1\}}{\arg\max}\big[\mathcal{I}(M)(b)\big],$$

and for all other symbols set $\alpha(\mathcal{I}) := \mathcal{I}$. This definition is *not* possible in the general probabilistic case, because probability measures often return zero on *all* single values. It is also a non-deterministic interpretation since it returns a *set* of values instead of a single value. The resulting semantics is three-valued, as in section 4.1. This arg max transformation is just one example of many possible NeSy transformations.

Then, in the practical implementation of ULLER, we use random sampling (see section 4.4) over a uniform distrubtion to obtain a single value from the set of values $\alpha(\mathcal{I})(m)$. This gives a precise foundation for the use of arg max in the classical semantics in [15].

## 6. Conclusion

The ULLER language [15] aims at a unifying foundation for neurosymbolic systems. In this paper, we have developed a new semantics for ULLER, based on Moggi's formalisation of computational effects as monads. In contrast to the original semantics, our semantics is truly modular. It is based on a notion of NeSy framework that provides the structure of the computational effects and the space of truth values. This modularity will enable a cleaner, more modular implementation of ULLER in Python and an easier integration of new frameworks, as well as a structured method of translating between different frameworks. First implementations of our mULLER framework are available in Python and Haskell, see https://github.com/cherryfunk/mULLER. Note that the distributional and probabilistic NeSy frameworks will make parameterized interpretations in the sense of [15] differentiable and that this can be integrated by using a category of differentiable manifolds and functions.

Our work suggests an analogy between ULLER's formulas $x := m(T_1, \ldots, T_n)(F)$ and Haskell's do-notation **do** $x \leftarrow m(T_1, \ldots, T_n); F$ for computational effects. Inspired by this analogy, one could extend ULLER to a language with computational terms and formulas that may be nested.

---

7. Or in case of an infinite distribution (see Appendix C.1)

8. For those interested in category theory; this is in fact a natural transformation, hence the name.

## Acknowledgments

We thank Rick Adamy for the idea to use monads in the context of ULLER, Kai-Uwe Kühnberger for initiating our collaboration, Emile van Krieken for useful discussions and Björn Gehrke for helping us with the implementation in Python. Also, big thanks to my (Daniel's) father for providing me the space-time to work on this paper and to Leilani H. Gilpin for some mental support.

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

## Appendix A. Brief Introduction to Category Theory

The main part of this paper does not use category theory. Indeed, we have introduced set-based notions of monad and of Double semigroup bounded lattice. They do not rely on category theory, nor do the central definitions of mULLER, in particular, the notions of NeSy framework, of signature, interpretation and formula, nor the rules of the Tarskian semantics.

By contrast, in this appendix, we will heavily make use of category theory. The main purpose of that is the possibility to work with continuous probability distributions.[9] Another motivation is the need for quantification structures for infinite domains, which are common in first-order logic. In these cases, we need to work with structured objects and structure-preserving maps, like measurable spaces and measurable functions. Without organising such spaces and maps into a category, we still can use the set-theoretic variant of

---

9. The lack of which is described as a major limitation and shortcoming of traditional NeSy systems by [13].

the semantics. However, we would need to restrict to structure-preserving maps in places like the definition of interpretation (Def. 6), and, more severely, would need to prove that the semantic rules in Def. 8 again yield structure-preserving maps.

When using category theory, we can avoid such proofs and base our theory on a certain structure that is required for the involved categories. That said, we still can use the set-theoretic semantics rules for convenience, also for categories other than **Set**, just because the categorical version of the rules tells us that the resulting maps will be structure-preserving.[10]

We now recall some basic notions of category theory. See [14] for an introduction with a focus on application in database theory, [2] for a logical and type-theoretic overview, and [7] as a general reference.

**Definition 10** *A Category* **C** *consists of*

- *a class $|\mathbf{C}|$ of objects,*

- *for any two objects $A, B \in |\mathbf{C}|$ a set $\mathbf{C}(A, B)$ of morphisms from $A$ to $B$. $f \in \mathbf{C}(A, B)$ is written as $f : A \to B$ (not necessarily a function),*

- *for any object $A \in |\mathbf{C}|$ an identity morphism $\mathrm{id}_A \in \mathbf{C}(A, A)$, i.e. $\mathrm{id}_A : A \to A$,*

- *for any $A, B, C \in |\mathbf{C}|$ a composition operation $\circ : \mathbf{C}(B, C) \times \mathbf{C}(A, B) \to \mathbf{C}(A, C)$, i.e. for $f : A \to B, g : B \to C$, we have $g \circ f : A \to C$,*

*such that*

- *identities are neutral elements for composition, i.e. $f \circ \mathrm{id}_A = f = \mathrm{id}_B \circ f$, and*

- *composition is associative, i.e. $(f \circ g) \circ h = f \circ (g \circ h)$.*

Examples of categories are:

- **Set**: Sets and functions.

  - $|\mathbf{Set}| = \{M \mid M \text{ is a set}\}$
  - $\mathbf{Set}(A, B) = \{f \mid f : A \to B \text{ is a function}\}$
  - Compositions and identities of functions.

- **Meas**: measurable spaces and measurable functions.[11]

  - $|\mathbf{Meas}| = \{(X, \Sigma_X) \mid \Sigma_X \text{ is a } \sigma\text{-algebra on } X\}$
  - $\mathbf{Meas}((X, \Sigma_X), (Y, \Sigma_Y)) = \{f : X \to Y \mid f^{-1}(B) \in \Sigma_X \text{ for all } B \in \Sigma_Y\}$
  - Compositions and identities of measurable functions.

- **Measr**: measure spaces and measure preserving functions.[12]

  - $|\mathbf{Measr}| = \{(X, \Sigma_X, \mu_X) \mid \mu_X \text{ is a measure on } (\Sigma_X, X)\}$

---

10. Technically, this holds for any construct (set-based category) in the sense of [1].

11. More details at https://ncatlab.org/nlab/show/measurable+space.

12. More details at https://ncatlab.org/nlab/show/measure+space.

- $\mathbf{Measr}((X, \Sigma_X, \mu_X), (Y, \Sigma_Y, \mu_Y)) = \{f : X \to Y \mid f \text{ is measure preserving}\}$
- Compositions and identities of measure preserving functions.

- **Prob**: probability spaces and measure preserving functions.[13]

  - $|\mathbf{Prob}| = \{(X, \Sigma_X, \rho_X) \mid \rho_X \text{ is a probability measure on } (\Sigma_X, X)\}$
  - $\mathbf{Prob}((X, \Sigma_X, \rho_X), (Y, \Sigma_Y, \rho_Y)) = \{f : X \to Y \mid f \text{ is measure preserving}\}$
  - Compositions and identities of measure preserving functions.

Commutative diagrams are often used to visualise equalities between (compositions of) morphisms in categories. For example, the following diagram shows the composition of morphisms $f : A \to B$ and $g : B \to C$:

$$
\begin{array}{ccc}
 & B & \\
f \nearrow & & \searrow g \\
A \xrightarrow{\ g \circ f\ } & & C
\end{array}
$$

**Definition 11** *An object* $1 \in |\mathbf{C}|$ *is called* terminal, *if for each* $A \in |\mathbf{C}|$ *there exists a unique morphism* $!_A : A \to 1_{\mathbf{C}}$.

Examples of terminal objects in **Set** are all singleton sets.

**Definition 12 (Products)** *Let* $\mathcal{C}$ *be a category and let* $A, B$ *be objects in* $\mathcal{C}$. *A* product *of* $A$ *and* $B$ *is an object* $A \times B$ *together with two morphisms (called projections)* $\pi_A : A \times B \to A$ *and* $\pi_B : A \times B \to B$ *such that for any object* $X$ *with morphisms* $f : X \to A$ *and* $g : X \to B$, *there exists a unique morphism* $u : X \to A \times B$ *such that* $\pi_A \circ u = f$ *and* $\pi_B \circ u = g$. *This unique* $u$ *is noted as* $\langle f, g \rangle$ *and is called the* pairing *of* $f$ *and* $g$.

*This universal property can be depicted by the following commutative diagram:*

$$
\begin{array}{ccc}
 & X & \\
f \swarrow & \Big\downarrow u = \langle f, g \rangle & \searrow g \\
A \xleftarrow{\ \pi_A\ } A \times B \xrightarrow{\ \pi_B\ } B
\end{array}
$$

This easily generalises to products of finitely many objects. A category having a terminal object and binary products is called *cartesian category.*

## Appendix B. Categorical NeSy Frameworks

### B.1. More (on) Monads

**Definition 13** *A **Kleisli triple** or **monad** on a category* $\mathcal{C}$ *involves a mapping of objects* $\mathcal{T} : \mathrm{Ob}(\mathcal{C}) \to \mathrm{Ob}(\mathcal{C})$, *a family of morphisms* $\eta_X : X \to \mathcal{T}X$ *for each object* $X$ *in* $\mathcal{C}$ *(called the* unit*), and a function that assigns to each morphism* $f : X \to \mathcal{T}Y$ *a morphism* $f^* : \mathcal{T}X \to \mathcal{T}Y$ *such that the axioms hold as in Def.* 1. *Note that a set-based Kleisli triple is then a Kleisli triple on the category* **Set**.

---

13. More details at https://ncatlab.org/nlab/show/Prob.

**Definition 14 (Strong Kleisli triple)** *A Kleisli triple $(\mathcal{T}, \eta, (-)^*)$ on a cartesian category $\mathcal{C}$ is called **strong** if there is a natural transformation*

$$\mathcal{S}_{A,B} : A \times \mathcal{T}B \longrightarrow \mathcal{T}(A \times B)$$

*satisfying the naturality condition: for any morphisms $f : A \to A'$ and $g : B \to B'$,*

$$\mathcal{T}(f \times g) \circ \mathcal{S}_{A,B} = \mathcal{S}_{A',B'} \circ (f \times \mathcal{T}g),$$

*and such that*

$$\mathcal{S}_{A,B} \circ (\mathrm{id}_A \times \eta_B) = \eta_{A \times B}, \qquad \mathcal{S}_{A,B} \circ (\mathrm{id}_A \times f^*) = (id \times f)^* \circ \mathcal{S}_{A,C}.$$

**Example 3** *Identity monad $\mathcal{I}$ on **Set**, the category of sets and functions. For a set $X$:*

$$\mathcal{I}X := X \quad \textit{(identity functor)},$$

$$\eta_X(x) := x, \quad f^*(x) := f(x) \quad \big(f : X \to Y, x \in X\big).$$

**Example 4** *Powerset monad $\mathcal{P}$ on **Set**, the category of sets and functions. For a set $X$:*

$$\mathcal{P}X := \{A \subseteq X\} \quad \textit{(powerset of $X$)},$$

$$\eta_X(x) := \{x\}, \quad f^*(A) := \bigcup_{a \in A} f(a) \quad \big(f : X \to \mathcal{P}Y, A \subseteq X\big).$$

**Example 5 (Sub-Distribution Monad $\mathcal{S}$)** *The sub-distribution monad $\mathcal{S}$ is similar to the distribution monad $\mathcal{D}$ but it allows for finitely supported measures that do not sum up to 1, that means:*

$$\mathcal{S}X := \Big\{ \rho : X \to [0,1] \ \Big| \ \sum_{x \in X} \rho(x) \leq 1, \ \rho \ \textit{countably additive and has finite support}\Big\},$$

$$\eta_X(x) := \delta_x, \delta_x(y) = \begin{cases} 1, & x = y \\ 0, & x \neq y \end{cases} \quad f^*(\rho)(y) := \sum_{x \in X} \rho(x)f(x)(y) \quad \big(f : X \to \mathcal{S}Y, \ \rho \in \mathcal{S}X\big).$$

**Example 6** *Giry monad $\mathcal{G}$ on **Meas**, the category of measurable spaces and maps. For a measurable space $(X, \Sigma_X)$ and dirac measure $\delta_x$ on $X$ for $x \in X$:*

$$\mathcal{G}(X, \Sigma_X) := \big\{\rho : X \to [0,1] \ \big| \ \rho(X) = 1, \ \rho \ \textit{countably additive}\big\} \quad \textit{(prob. measures on $X$)},$$

$$\eta_{(X,\Sigma_X)}(x) := \delta_x, \quad \delta_x(A) = \begin{cases} 1, & x \in A \\ 0, & x \notin A \end{cases}$$

$$f^*(\rho)(A) := \int_X f(x)(A)d\rho(x) \quad \big(f : X \to \mathcal{G}Y, A \subseteq Y \ \textit{measurable}\big).$$

*$\delta_x$ is the probability distribution that assigns all probability mass to $x$.*

**Example 7** *Infinite Giry monad $\mathcal{G}_\infty$ on* **Meas.** *For a measurable space $(X, \Sigma_X)$:*

$$\mathcal{G}_\infty(X, \Sigma_X) := \big\{ \mu : \Sigma_X \to [-\infty, \infty] \mid \mu(\emptyset) = 0, \ \mu \ countably \ additive \big\},$$

$$\eta_{(X,\Sigma_X)}(x) := \delta_x, \qquad \delta_x(A) = \begin{cases} 1 & x \in A \\ 0 & x \notin A \end{cases},$$

$$f^*(\mu)(A) := \int_X f(x)(A) \, d\mu(x) \quad \big( f : X \to \mathcal{S}Y, \ A \subseteq Y \ measurable \big).$$

*Here the integral is the Lebesgue–Stieltjes integral with respect to the extended signed measure $\mu$. Writing the Jordan decomposition $\mu = \mu^+ - \mu^-$ and using linearity of the integral, one checks that the monad laws hold; thus $\mathcal{S}$ extends the Giry monad by allowing negative and (possibly) infinite total mass.*

**Proposition 15 (and definition of the measure-space monad $\mathcal{M}$)** *We can define a measure monad $\mathcal{M}$ as a monad $(\mathcal{T}, \eta, \mu)$[14] on the category* **Measr** *of measure spaces with $\eta, \mu$ being the unit and multiplication of the Giry monad $\mathcal{G}$ on* **Meas**. *For $\rho$ being probability measures, we can define a probability-space monad $\mathcal{O}$ on the category* **Prob** *of probability spaces. The same construction can be applied to obtain an infinite measure-space monad $\mathcal{M}_\infty$ on* **Measr**.

$$\mathcal{M}((X, \rho)) := (\mathcal{G}(X), \rho^\eta),$$
$$\rho^\eta := B \longmapsto \rho(\eta^{-1}(B)), \ for \ B \subseteq \mathcal{G}(X) \ measurable,$$
$$\eta^{\mathcal{M}} := \eta, \quad \mu^{\mathcal{M}} := \mu.$$

**Proof** If $\rho$ is a probability measure, we know that $\rho^\eta(\mathcal{G}(X)) = \rho(\eta^{-1}(\mathcal{G}(X))) = \rho(X) = 1$, countable additivity follows alike. Now we only need to check whether $\eta^{\mathcal{M}}, \mu^{\mathcal{M}}$ are measure preserving, which follows for the unit by definition:

$$\eta^{\mathcal{M}} : (X, \rho) \longrightarrow \big( \mathcal{G}(X), \rho^\eta \big)$$

$$\rho\big( (\eta^{\mathcal{M}})^{-1}(B) \big) = \rho\big( \eta^{-1}(B) \big) = \rho^\eta(B).$$

Now, for the multiplication we have:

$$\mu^{\mathcal{M}} : \big( \mathcal{G}^2(X), \rho \circ \eta^{-1} \circ \eta_{\mathcal{G}}^{-1} \big) \longrightarrow \big( \mathcal{G}(X), \rho \circ \eta^{-1} \big)$$

since

$$M^2(X) = M\big( (\mathcal{G}(X), \rho^\eta) \big) = \big( \mathcal{G}^2(X), (\rho^\eta)^\eta \big),$$

and that means we can write for any measurable set $A \subseteq \mathcal{G}^2(X)$:

$$(\rho^\eta)^\eta(A) = \rho^\eta\big( \eta_{\mathcal{G}}^{-1}(A) \big) = \rho\big( \eta^{-1} \circ \eta_{\mathcal{G}}^{-1}(A) \big).$$

---

14. Here we use the traditional category theoretical definition of monad as $(\mathcal{T}, \eta, \mu)$, where $\mu$ is monad multiplication. This is however equivalent to the definition as Kleisli triple.

Therefore we show the measure-preserving property as follows

$$\rho \circ \eta \circ \eta_{\mathcal{G}}^{-1}\big(\mu^{-1}(B)\big) \;=\; \rho \circ \eta^{-1}\big(\eta_{\mathcal{G}}^{-1}(\mu^{-1}(B))\big) \;=\; \rho \circ \eta^{-1}(B),$$

because we know the following fact from the monad laws:

$$A = \eta_{\mathcal{G}}^{-1}\big(\mu^{-1}(B)\big) \;\Longleftrightarrow\; A = \mu\big(\eta_{\mathcal{G}}(A)\big) = B.$$

∎

**B.2. The 2Sg-Bl Algebra: A Comprehensive Overview**

In this subsection, we provide a comprehensive overview of the different algebraic structures that can serve as algebras on the truth space $\mathcal{T}\Omega$ in our neurosymbolic framework, along with their associated operations types and logical properties.

Table B.2 provides a comprehensive comparison of the fundamental operations across different algebraic structures, showing how each algebra defines its basic operations.

| Algebra | Set | $\perp$ | $\top$ | $\oplus$ | $\otimes$ | $\rightarrow$ | $\neg$ | aggr$^\exists$ |
|---|---|---|---|---|---|---|---|---|
| Boolean | $\{0,1\}$ | 0 | 1 | max | min | $I_B$ | $\neg_R$ | sup |
| LTN$_p$ | $[0,1]$ | 0 | 1 | $S_P$ | $T_P$ | $I_{SP}$ | $\neg_C$ | $\lVert \cdot \rVert_p$ |
| LTN$_q$ | $[0,1]$ | 0 | 1 | $S_P$ | $T_P$ | $I_{SP}$ | $\neg_C$ | $P\exists_q$ |
| Product | $[0,1]$ | 0 | 1 | $S_P$ | $T_P$ | $I_P$ | $\neg_R$ | $P\exists$ |
| S-Prod. | $[0,1]$ | 0 | 1 | $S_P$ | $T_P$ | $I_{SP}$ | $\neg_C$ | $P\exists$ |
| Priest | $\{F,B,T\}$ | $F$ | $T$ | max | min | $I_{KD}$ | $\neg_C$ | sup |

Table 3: Overview of some aggregated 2Sg-Bl with named operations

**t-conorms and t-norms ($\oplus$ and $\otimes$):**

- $S_P$ (Probabilistic sum): $xS_Py = x + y - xy$

- $T_P$ (Product): $xT_Py = xy$

**Implications ($\rightarrow$):**

- $I_P$ (Product/Goguen): $I_P(x,y) = \begin{cases} 1 & \text{if } x \leq y \\ y/x & \text{otherwise} \end{cases}$

- $I_B$ (Boolean): $I_B(x,y) = \begin{cases} 0 & \text{if } x = 1, y = 0 \\ 1 & \text{otherwise} \end{cases}$

- $I_S$ (General S-implication): $I_S(x,y) = \neg x \oplus y$

- $I_{KD}$ (Kleene-Dienes/Material): $I_{KD}(x,y) = \max(1-x, y)$

- $I_{SP}$ (S-Product): $I_{SP}(x,y) = 1 - x + xy$

**Negations ($\neg$):**

- $\neg_R$ (Residual): $\neg_R x = x \to 0$ (includes Heyting/intuitionistic negation)

- $\neg_C$ (Classical/1-Involutive): $\neg_C x = 1 - x$

- $\neg_V$ (0-Involutive): $\neg_V x = 0 - x = -x$

**Aggregations ($\mathrm{aggr}^\exists$):**

- $P\exists$ (Infinitary Probabilistic Sum): $P\exists(x) = 1 - \exp\!\Big(\mathbb{E}_{a\sim\mu}\Big[1 - \ln[\![F]\!]_{\nu_F[x\mapsto a]}\Big]\Big)$

- $P\exists_q$ (($\mu, q$)-approximated $P\exists$): $P\exists_q(x) = 1 - \exp\!\Big(\mathbb{E}_{a\sim\mu}\Big[\big(1 - \ln[\![F]\!]_{\nu_F[x\mapsto a]}\big)^q\Big]^{\frac{1}{q}}\Big)$
  for $1/2 \le q \le 1$, with $P\exists_q \to P\exists$ as $q \to 1$. Here $\mu$ can be any measure and it depends on the context, in $\mathrm{LTN}_q$ it depends on the measure space of the sort of the quantified variable at hand.

The quantification aggregations employ logarithmic and exponential transforms because they provide the natural generalization of the product (or probabilistic sum) to infinitary domains. While finite probabilistic sums can be computed directly using products, extending to infinite domains requires the use of expectations, and the logarithmic and exponential transforms enable this generalization while preserving the essential structure of probabilistic aggregation.

### B.3. Definition of Categorical NeSy Frameworks

**Definition 16 (Internal Aggregated 2Sg-Bl)** *An aggregated 2Sg-Bl internal to a cartesian category $\mathcal{C}$ on an object $A$ in $\mathcal{C}$ consists of a lattice on $A$ internal[15] to $\mathcal{C}$, morphisms $\oplus, \otimes, \to\colon A \times A \to A$, $\perp, \top : 1_{\mathcal{C}} \to A$ and for any objects $B$ and $C$ maps $\mathrm{aggr}^\vee_{B,C}, \mathrm{aggr}^\exists_{B,C} : \mathcal{C}(B \times C, A) \to \mathcal{C}(B, A)$, such that the axioms of Def. 2 hold when appropriately interpreted in $\mathcal{C}$[16].*

Note that our categorical handling of aggregation differs from that in the set-theoretic setting. A full analogy to the set-theoretic case would require aggregation morphisms $A^X \to X$, which would need a cartesian closed category. This requirement seems too strong for our purposes, since it is not met in many examples and only is required for interpreting higher-order logics. However, given a cartesian closed category (like **Set**) with aggregation $aggr : A^X \to X$, we can define aggregation in the sense of Def. 16 as mapping $f : B \times C \to A$ to $B \xrightarrow{\Lambda(f)} A^C \xrightarrow{aggr} A$. Here, $\Lambda(f)$ is currying, defined as follows in **Set**: $\Lambda(f)(x)(y) = f(x, y)$. In the sequel, we will rely on this definition also for **Set**-based categories (constructs [1]) that are not cartesian closed, noting that the definition works even if $A^C$ is just a set and not an object in the category. Hence, in examples, we will define aggregation as in Def. 2.

**Definition 17 (NeSy framework)** *A NeSy framework $(\mathcal{T}, \Omega, \mathcal{R})$ consists of*

---

15. For an explanation check https://ncatlab.org/nlab/show/internalization, since this is out of scope for this paper.
16. $1_{\mathcal{C}}$ is a terminal object.

1. *a strong monad $\mathcal{T}$ with strength $\mathcal{S}$ on a cartesian category $\mathcal{C}$,*[17]

2. *An object $\Omega \in \mathrm{Ob}_{\mathcal{C}}$ acting as truth basis,*[18]

3. *an aggregated 2Sg-Bl $\mathcal{R}$ internal to $\mathcal{C}$ on the truth space $\mathcal{T}\Omega$.*

Examples are given in Table 4 and their semantics are discussed in section C. Note that further examples arise by varying the 2Sg-Bl $\mathcal{R}$ on $[0, 1]$.

Table 4: NeSy Framework Examples (categorical)

| Logic/Theory | $\mathcal{C}$ | $\mathcal{T}$ | $\Omega$ | $\mathcal{T}\Omega$ | $\mathcal{R}$ | Sem. |
|---|---|---|---|---|---|---|
| Probabilistic | **Measr** | Measure-space $\mathcal{M}$ | $\{0,1\}$ | $[0,1]$ | Product BL–Alg. | §C.1 |
| Infinitary LTN$_p$ | **Prob** | Probability-space $\mathcal{O}$ | $\{0,1\}$ | $[0,1]$ | Product SBL-Alg. | §C.2 |
| STL$_r$ | **Measr** | $\infty$-measure-space $\mathcal{M}_\infty$ | $\{1\}$ | $[-\infty,\infty]$ | approx. $\mathbb{R}$ | §C.3 |

## Appendix C. Categorical Semantics

The categorical notion of interpretation differs from the set-theoretic definition (Definition 6) only in that *sets* are replaced by *objects* in the category $\mathcal{C}$ and *functions* are replaced by *morphisms* in $\mathcal{C}$. This generalization allows the framework to work in any category with suitable structure, not just the category of sets and functions.

**Definition 18 (Tarskian semantics $[\![\cdot]\!]$ of formulas)** *Given a NeSy framework $(\mathcal{T}, \Omega, \mathcal{R})$ and a NeSy interpretation $\mathcal{I}$ we can determine the interpretation morphisms:*

$$\textbf{Formulas: } [\![F]\!]_{\mathcal{I}} : \mathcal{V}_F \to \mathcal{T}\Omega, \quad \textbf{Terms: } [\![T]\!]_{\mathcal{I}} : \mathcal{V}_T \to \mathcal{I}(s_T) :$$

**Remark 19** *We define $\mathcal{V}_T := \prod_{x:t\in\Gamma_T} \mathcal{I}(t)$. Here $\Gamma_T$ is the context of $T$ and $s_T$ is the (unique) sort of the term $T$. Analogously $\mathcal{V}_F := \prod_{x:t\in\Gamma_F} \mathcal{I}(t)$. Note that if $T_1$ is a subterm of $T_2$, there is a projection $\pi_{T_1,T_2} : \mathcal{V}_{T_2} \to \mathcal{V}_{T_1}$, and analogously for formulas. $\textbf{T}$ stands for $T_1, \ldots, T_n$. Moreover, $\langle [\![\textbf{T}]\!]_i \circ \pi_i \rangle_i = \langle [\![T_1]\!] \circ \pi_1, \ldots, [\![T_n]\!] \circ \pi_n \rangle$ and $[\![\textbf{T}]\!] = ([\![T_1]\!], \ldots, [\![T_n]\!])$. The categorical semantics ensures that all involved and resulting functions are morphisms in $\mathcal{C}$, i.e. are measurable in case that $\mathcal{C} = \textbf{Meas}$, etc. With a purely set-theoretic semantics, we would need to prove measurability (or other properties) separately for each NeSy framework. That said, besides the general categorical case, [19] for better understandability, we also translate the equations to their meaning in the category of sets. We work with variable valuations $\nu \in \mathcal{V}_T$ (and $\nu \in \mathcal{V}_F$), noting that elements of $\prod_{x:t\in\Gamma_T} \mathcal{I}(t)$ map variables $x : t$ to values in $\mathcal{I}(t)$. We write $[\![T]\!]_{\mathcal{I},\nu} = [\![T]\!]_{\mathcal{I}}(\nu)$ and $[\![F]\!]_{\mathcal{I},\nu} = [\![F]\!]_{\mathcal{I}}(\nu)$. That said, we mostly omit $\mathcal{I}$ and $\nu$ if clear from the context.*

---

17. Note that **Set** is a cartesian closed category, and every monad on **Set** is strong.

18. Similar to the basis of a vector space.

19. Logician's note: We don't differentiate properly between additive and multiplicative connectives/units/quantifiers for an easier presentation coherent with the NeSy literature. However, this could easily be adapted to obtain something like Girad's linear logic [5], following the Zeitgeist of his transcendental syntax.

Table 5: Inductive definition of the Tarskian semantics

| Syntax | Categorical Semantics $[\![\cdot]\!]_{\mathcal{I}}$ | Set Semantics $[\![\cdot]\!]_{\mathcal{I},\nu}$ |
|---|---|---|
| **Terms** | | |
| $[\![x:s]\!]$ | $\mathrm{id}_{\mathcal{I}(s)}$ | $\nu_s(x)$ |
| $[\![c]\!]$ | $\mathcal{I}(c)$ | $\mathcal{I}(c)$ |
| $[\![T.\mathsf{prop}]\!]$ | $\mathcal{I}(\mathsf{prop}) \circ [\![T]\!]$ | $\mathcal{I}(\mathsf{prop})([\![T]\!])$ |
| $[\![f(\boldsymbol{T})]\!]$ | $\mathcal{I}(f) \circ \langle [\![\boldsymbol{T}]\!]_i \circ \pi_i \rangle_i$ | $\mathcal{I}(f)([\![\boldsymbol{T}]\!])$ |
| **Atomic formulas** | | |
| $[\![P]\!]$ | $\eta_\Omega \circ \mathcal{I}(P)$ | $\eta_\Omega(\mathcal{I}(P))$ |
| $[\![N]\!]$ | $\mathcal{I}(N)$ | $\mathcal{I}(N)$ |
| $[\![R(\boldsymbol{T})]\!]$ | $\eta_\Omega \circ \mathcal{I}(R) \circ \langle [\![\boldsymbol{T}]\!]_i \circ \pi_i \rangle_i$ | $\eta_\Omega\big(\mathcal{I}(R)([\![\boldsymbol{T}]\!])\big)$ |
| $[\![M(\boldsymbol{T})]\!]$ | $\mathcal{I}(M) \circ \langle [\![\boldsymbol{T}]\!]_i \circ \pi_i \rangle_i$ | $\mathcal{I}(M)([\![\boldsymbol{T}]\!])$ |
| **Compound formulas** | | |
| $[\![\top]\!], [\![\bot]\!]$ | $1_\mathcal{R}, 0_\mathcal{R}$ | $1_\mathcal{R}, 0_\mathcal{R}$ |
| $[\![\neg F]\!]$ | $\neg_\mathcal{R} \circ [\![F]\!]$ | $\neg_\mathcal{R}([\![F]\!])$ |
| $[\![F \to G]\!]$ | $\to_\mathcal{R} \circ \langle [\![F]\!] \circ \pi_F, [\![G]\!] \circ \pi_G \rangle$ | $[\![F]\!] \to_\mathcal{R} [\![G]\!]$ |
| $[\![F \| G]\!]$ | $\oplus_\mathcal{R} \circ \langle [\![F]\!] \circ \pi_F, [\![G]\!] \circ \pi_G \rangle$ | $[\![F]\!] \oplus_\mathcal{R} [\![G]\!]$ |
| $[\![F \& G]\!]$ | $\otimes_\mathcal{R} \circ \langle [\![F]\!] \circ \pi_F, [\![G]\!] \circ \pi_G \rangle$ | $[\![F]\!] \otimes_\mathcal{R} [\![G]\!]$ |
| $[\![\exists x{:}s\, F]\!]$ | $\mathrm{aggr}^{\exists}_{V_{F \setminus x:s}, \mathcal{I}(s)}([\![F]\!])$ | $\mathrm{aggr}^{\exists}_{\mathcal{I}(s)}(\lambda a.[\![F]\!]_{\nu_F[x \mapsto a]})$ |
| $[\![\forall x{:}s\, F]\!]$ | $\mathrm{aggr}^{\forall}_{V_{F \setminus x:s}, \mathcal{I}(s)}([\![F]\!])$ | $\mathrm{aggr}^{\forall}_{\mathcal{I}(s)}(\lambda a.[\![F]\!]_{\nu_F[x \mapsto a]})$ |
| $[\![x := m(\boldsymbol{T})(F)]\!]$ | $[\![F]\!]^* \circ \mathcal{S} \circ \big\langle \pi_{V_{F \setminus x:s}}, \mathcal{I}(m) \circ \langle [\![\boldsymbol{T}]\!]_i \circ \pi_i \rangle_i \big\rangle$ | $\mathbf{do}\ a \leftarrow \mathcal{I}(m)([\![\boldsymbol{T}]\!]);\ [\![F]\!]_{\nu[x \mapsto a]}$ |

## C.1. Probabilistic Semantics

**Definition 20** *A **probabilistic NeSy framework** is a tuple $(\mathcal{T}, \Omega, \mathcal{R})$ with $\mathcal{T} = \mathcal{M}$ the Giry monad on the category **Measr** of measure spaces and $\Omega$ the truth basis, normally $\Omega = \{0, 1\}$, and $\mathcal{R}$ a suitable 2Sg-Bl, for example the Product BL-Algebra.*

The interpretation of a function $f$ of arity $n$ is a (measure preserving) *Markov kernel*, which is a measurable map $X \xrightarrow{q} \mathcal{M}(Y)$ where $\mathcal{M}$ denotes the *measure monad* on the category **Measr** of measure spaces.

Our definition of a probabilistic semantics largely follows that in the original ULLER paper [15]. A central design decision of ULLER is the use of first-order interpretations and the use of probability distributions to interpret computational function symbols. This means that ULLER (and therefore also mULLER) is (like Logic Tensor Networks) not built on weighted model counting, i.e. on probability distributions over the set of interpretations.

That said, it is still possible to capture certain aspects of weighted model counting in ULLER and mULLER, as we will see in section C.4 below.

Connectives in the probabilistic semantics of ULLER are interpreted assuming independence of probabilities for atomic formulas. Hence, our distributional semantics can be seen as a special case of a fuzzy semantics, where the t-norm is the probabilistic product and the t-conorm is the probabilistic sum. This means that we can use the same equations as in the fuzzy semantics (and as in LTNs), but with a different motivation. Moreover, this explains why there is no essential difference between these probabilistic and fuzzy semantics.

Let us derive from our general semantic in definition 8 the interpretation of monadic formulas in probabilistic semantics. For the probabilistic NeSy framework, we define the aggregation morphisms $\mathrm{aggr}^{\forall}_{B,C}, \mathrm{aggr}^{\exists}_{B,C}$ required by Def. 16 as follows: for any measure spaces $B$ and $C$,

$$\mathrm{aggr}^{\forall}_{B,C}(f) := \exp \circ \mathbb{E}_{c \sim \mu_C}[\ln \circ f(\cdot, c)]$$
$$\mathrm{aggr}^{\exists}_{B,C}(f) := (1 - \exp) \circ \mathbb{E}_{c \sim \mu_C}[1 - \ln \circ f(\cdot, c)]$$

where $f : B \times C \to [0, 1]$ and $\mu_C$ is the measure on $C$. In the following examples, we provide implicit definitions of these aggregation operations through their concrete realizations. In set-theoretic notation, the semantics of computational formulas can be written as, where $\rho_m(\cdot|\boldsymbol{T}) := \mathcal{I}(m)(\llbracket \boldsymbol{T} \rrbracket)$:

$$
\begin{aligned}
\llbracket x := m(\boldsymbol{T})(F) \rrbracket &= \textbf{do} \ \ a \leftarrow \mathcal{I}(m)(\llbracket \boldsymbol{T} \rrbracket); \ \llbracket F \rrbracket_{\nu[x \mapsto a]} \\
&= \int_{a \in \mathcal{I}(s_m)} \llbracket F \rrbracket_{\nu_F[x \mapsto a]} \, d\mathcal{I}(m)(\llbracket \boldsymbol{T} \rrbracket)(a) \\
&= \int_{a \in \mathcal{I}(s_m)} \llbracket F \rrbracket_{\nu_F[x \mapsto a]} \, d\rho_m(a|\boldsymbol{T}) \\
&= \mathbb{E}_{a \sim \rho_m(\cdot|\boldsymbol{T})} \big[ \llbracket F \rrbracket_{\nu[x \mapsto a]} \big] \\
&= \sum_{a \in \mathcal{I}(s_m)} \llbracket F \rrbracket_{\nu_F[x \mapsto a]} \cdot \rho_m(a|\boldsymbol{T}) \quad (\text{if } \mathcal{I}(s_m) \text{ is finite})
\end{aligned}
$$

We evaluate in the Product Algebra to obtain, where $\mu_s$ is the measure given by the measure space of $\mathcal{I}(s)$

$$\llbracket x := m(\boldsymbol{T})(F) \rrbracket := \mathbb{E}_{a \sim \rho_m(\cdot|\boldsymbol{T})} \big[ \llbracket F \rrbracket_{\nu[x \mapsto a]} \big] \tag{14}$$

$$\llbracket \exists x{:}s \ F \rrbracket := 1 - \exp\Big( \mathbb{E}_{a \sim \mu_s} \big[ 1 - \ln \llbracket F \rrbracket_{\nu_F[x \mapsto a]} \big] \Big) \tag{15}$$

$$\llbracket \forall x{:}s \ F \rrbracket = \exp \mathbb{E}_{a \sim \mu_s} \big[ \ln \llbracket F \rrbracket_{\nu[x \mapsto a]} \big] \overset{(\text{finite, } \mu_s \text{ avrg. count. meas.})}{=} \prod_{a \in \mathcal{I}(s)} \llbracket F \rrbracket_{\nu[x \mapsto a]} \tag{16}$$

$$\llbracket F \| G \rrbracket := \llbracket F \rrbracket + \llbracket G \rrbracket - \llbracket F \rrbracket \cdot \llbracket G \rrbracket, \quad \llbracket F \& G \rrbracket := \llbracket F \rrbracket \cdot \llbracket G \rrbracket, \tag{17}$$

$$\llbracket F \to G \rrbracket := \max\big(1, \llbracket G \rrbracket / \llbracket F \rrbracket\big), \quad \llbracket \neg F \rrbracket := 1 - \llbracket F \rrbracket \tag{18}$$

$$\llbracket \bot \rrbracket := 0, \quad \llbracket \top \rrbracket := 1. \tag{19}$$

## C.2. Infinitary LTN$_p$ Semantics

**Definition 21** *A **LTN-like NeSy framework** is a tuple $(\mathcal{T}, \Omega, \mathcal{R})$ with $\mathcal{T} = \mathcal{O}$ the probability monad on the category **Prob** of probability spaces and $\Omega$ the truth basis, normally $\Omega = \{0, 1\}$, and $\mathcal{R}$ a suitable 2Sg-Bl, for example the Product Real Algebra from Product Real Logic as in [3].*

**Setting** Stable product real logic of Logic Tensor Networks [3] uses $p$-means for finite quantification. The hyperparameter $p$ is usually increased during training, because this moves from mean (tolerant to outliers) towards the maximum[20] (logically stricter). However, since domains are generally infinite, we also need to aggregate infinite many truth–scores $(x_i)_{i \in I} \subseteq [0, 1]$. The power–mean extends from the finite case to an *integral* form that is well defined whenever the data are $L^p$-integrable. Let $(X, \mathcal{A}, \rho)$ be a probability space and $f : X \to [0, 1] \subseteq \mathbb{R}$ a measurable map with $\int_X f \, d\rho \leq 1$. Because $0 \leq f \leq 1$, one automatically has $f \in L^p(\rho)$ for every real $p$, so $p$-means are always defined. This bounded–by–one assumption reflects the fact that in our logical reading a truth-score never exceeds 1.

The infinitary LTN$_p$ semantics is a modification of the probabilistic semantics, which is motivated by replacing the quantifiers in equation (15) and (16). We generalize this: by working in the category **Prob**, for any sort $s$ we have to provide a probability measure $\rho_s$ on $\mathcal{I}(s)$. This enables us to obtain a $p$-means for infinite[21] domains:

$$M_p(a_1, \ldots, a_n) := \Big(\frac{1}{n} \sum_{i=1}^{n} a_i^p\Big)^{1/p}, \quad M_p(f; \rho_s) := \Big(\int_{x \in X} f(x)^p \, d\rho_s(x)\Big)^{1/p}, \qquad (20)$$

and these extend to $M_0(a_1, \ldots, a_n) := \Big(\prod_{i=1}^{n} a_i\Big)^{\frac{1}{n}}$ and $M_0(f; \rho_s) := \exp\big(\int \ln f \, d\rho_s\big)$. For $p \to \infty$ we recover the supremum. Take $X = \{1, \ldots, N\}$ with counting measure $1/N$, then $M_p(f; \rho_s)$ reduces to $M_p(a_1, \ldots, a_n)$ or choose weights $w_i$ summing up to 1 for $i = 1, \ldots, N$ to obtain the weighted $p$-mean. The aggregated 2Sg-Bl is similar to the probabilistic one, except for the Reichenbach implication as implication and the following aggregation functions. For a hyperparameter $1 \leq p < \infty$ of LTN$_p$, let $\mathrm{aggr}^{\exists}_{\mathcal{I}(s)}(f) := M_p(f; \rho_s)$ and $\mathrm{aggr}^{\forall}_{\mathcal{I}(s)}(f) := 1 - M_p(\lambda x. f(1 - x); \rho_s)$. As a result, equation (20) now becomes:

$$\llbracket \exists x{:}s \ F \rrbracket = \Big(\int_{a \in \mathcal{I}(s)} \big(\llbracket F \rrbracket_{\nu[x \mapsto a]}\big)^p \, d\rho_s(a)\Big)^{1/p}.$$

As in [3], this is for $1 \leq p < \infty$, where for $p \to \infty$ we recover the supremum. However, we also propose a different pair of quantifiers with $1/2 \leq q \leq 1$, where for $q \to 1$ the universal quantifier converges to the product, while the existential quantifier converges to

---

20. This holds for existential quantification. Universal quantification $\forall$ is defined through $\neg \exists x{:}s \neg$, and converges to the minimum.

21. This defends the idea of LTN against the criticism of [12], that the domains are finite and therefore no probability distribution is approximated.

the probabilistic sum:

$$[\![\forall x{:}s\ F]\!] := \exp\!\Big(\Big(\int_{a\in\mathcal{I}(s)}\big(\ln[\![F]\!]_{\nu[x\mapsto a]}\big)^q\,d\rho_s(a)\Big)^{1/q}\Big), \quad [\![\exists x{:}s\ F]\!] := [\![\neg\forall x{:}s\ \neg F]\!].$$

It is worth noting that our probability measure $\rho_s$ depend on the sort $s$ of the variable $x$ in the quantifier, since it is given by the probability measure of the probability space of $\mathcal{I}(s)$. This stands in contrast to [12], where the probability measure depends directly on the variable $x$.

## C.3. Infinitary STL Semantics

Signal Temporal Logic (STL) is a temporal logic for expressing properties of signals. STL is particularly useful for modeling and analyzing the temporal aspects of real-time systems, such as the timing and sequencing of events.

In the semantics of STL, we do not have any implication connective, nor neutral elements. We still need to interpret the syntactic implication connective as some form of semantical implication and the syntactic $\perp$ and $\top$ as $-\infty$ and $\infty$, the latter as in [12]. Also keep in mind, that this does not touch the truth designations of $-\infty$ as absolute falsity and $\infty$ as absolute truth.

What also can not be ignored is that STL does not directly use a 2Sg-Bl, but only approximates one. It works within the normal extended real numbers algebra $(\tilde{\mathbb{R}}, \max, \min, +, *)$ and then goes on to approximate the min and max operations. The are many different ways to do this, but one of the most recent ones is to use the $\mathsf{A}^r$ and $\mathsf{O}^r$ operators as defined in [17]. Additionally, STL is not concerned with the operations $\otimes$ and $\oplus$ of the 2Sg-Bl, these are not used in the semantics of STL, and are just kept to be the standard operations $+, *$ of the extended real numbers algebra.

For these reasons, in order to faithfully model STL, in a way that makes it comparable to other semantics, we would need to extend our syntax and semantics, and specifically, we would need to allow to approximate 2Sg-Bls. This however, is out of scope for this paper, and will be discussed in a future work, and yet we still give a first sketch:

$$[\![x := m(\boldsymbol{T})(F)]\!] := \mathbb{E}_{a\sim\mu_m(\cdot|\boldsymbol{T})}\big[[\![F]\!]_{\nu[x\mapsto a]}\big] \tag{21}$$

$$[\![\exists x{:}s\ F]\!] := \mathsf{O}^r_{a\in\mathcal{I}(s)}\big([\![F]\!]_{\nu[x\mapsto a]}\big), \quad [\![\forall x{:}s\ F]\!] := \mathsf{A}^r_{a\in\mathcal{I}(s)}\big([\![F]\!]_{\nu[x\mapsto a]}\big), \tag{22}$$

$$[\![F\|G]\!] := \mathsf{O}^r([\![F]\!], [\![G]\!]), \quad [\![F\&G]\!] := \mathsf{A}^r([\![F]\!], [\![G]\!]) \tag{23}$$

$$[\![F\to G]\!] := [\![\neg F\|G]\!] = \mathsf{O}(-[\![F]\!], [\![G]\!]), \quad [\![\neg F]\!] := -[\![F]\!] \tag{24}$$

$$[\![\perp]\!] := -\infty, \quad [\![\top]\!] := \infty. \tag{25}$$

The STL robustness metrics are defined as in [12] and originally in [17]:

$$\mathsf{A}^r_{a\in M}(a) = \begin{cases} \frac{\sum_a a_{min}e^{\tilde{a}}e^{r\tilde{a}}}{\sum_a e^{r\tilde{a}}} & \text{if } a_{min} < 0 \\[2mm] \frac{\sum_a a e^{-r\tilde{a}}}{\sum_a e^{-r\tilde{a}}} & \text{if } a_{min} > 0 \\[2mm] 0 & \text{if } a_{min} = 0 \end{cases}$$

where $r \in \mathbb{R}^+$ (constant), $a_{min} = \min_{a \in M}(a)$, and $\tilde{a} = \frac{a - a_{min}}{a_{min}}$. $\mathsf{A}^r$ is an approximation of the min operation, and for $r \to \infty$ it converges to it. Therefore, its notation is similar to the notation of the min operation, with $\mathsf{A}^r_{b \in N}(f(b)) := \mathsf{A}^r(\mathrm{im}(f)) := \mathsf{A}^r_{a \in \mathrm{im}(f)}(a)$. The operator $\mathsf{O}^r_{a \in M}$ is defined as $-\mathsf{A}^r_{a \in M}(-a)$.[22] For infinite domains, the minimum is replaced by the infimum $\inf_{a \in M}(a)$, and the summations $\sum_{a \in M}$ are replaced by integrals $\int_{a \in M} d\mu_s(a)$, where $\mu_s$ is the measure given by the measure space of $\mathcal{I}(s)$.

## C.4. Weighted Model Counting and Weighted Model Integration

ULLER can model certain aspects of weighted model counting (WMC) in a probabilistic semantics. However, instead of summing up literal or model weights, one needs to sum up weights of variable valuations. In the case of ULLER [15], we have the following definition. Given an interpretation $\mathcal{I}$ and a formula $F$ that is classical (i.e. without computational symbols) with context $\Gamma_F := \{x_1 : s_1, \ldots, x_n : s_n\}$, the domains of the variables are given by $\mathcal{I}(s_1), \ldots, \mathcal{I}(s_n)$[23]. This yields the weighted model count (WMC) as follows:

$$
\begin{aligned}
\mathrm{WMC}(F, w) &= \sum_{\boldsymbol{a} \in \mathcal{I}(s_1) \times \cdots \times \mathcal{I}(s_n)} w(\boldsymbol{a}) \llbracket F \rrbracket_{\nu[x_1 \mapsto a_1, \ldots, x_n \mapsto a_n]} \\
&= \sum_{a_1 \in \mathcal{I}(s_1)} \cdots \sum_{a_n \in \mathcal{I}(s_n)} w(a_1, \ldots, a_n) \llbracket F \rrbracket_{\nu[x_1 \mapsto a_1, \ldots, x_n \mapsto a_n]}
\end{aligned}
$$

If the weight function factorises—i.e. the random variables $x_1, \ldots, x_n$ are assumed **independent**[24]—then for every assignment $(a_1, \ldots, a_n) \in \mathcal{I}(s_1) \times \cdots \times \mathcal{I}(s_n)$:

$$
w(a_1, \ldots, a_n) = \prod_{i=1}^{n} \rho_{f_i}(a_i).
$$

Consequently, the weighted model count becomes

$$
\mathrm{WMC}(F, f_1, \ldots, f_n) = \sum_{a_1 \in \mathcal{I}(s_1)} \cdots \sum_{a_n \in \mathcal{I}(s_n)} \left( \prod_{i=1}^{n} \rho_{f_i}(a_i) \right) \llbracket F \rrbracket_{\nu[x_1 \mapsto a_1, \ldots, x_n \mapsto a_n]}.
$$

This WMC can be expressed in the language of NeSy systems as follows ([15], p. 234):

$$
x_1 := f_1(), \ldots, x_n := f_n()(F)
$$

In the linearly **dependent** case, rewrite it via the chain rule,

$$
w(a_1, \ldots, a_n) = \prod_{i=1}^{n} \rho_{f_i}(a_i \mid a_1, \ldots, a_{i-1}),
$$

---

22. See [17] for details.
23. In the original ULLER paper these were written as $\Omega_1 := \mathcal{I}(s_1), \ldots, \Omega_n := \mathcal{I}(s_n)$.
24. As in [15] (p. 16).

to make the conditional dependencies explicit. In this case, the WMC becomes

$$\sum_{a_1 \in \mathcal{I}(s_1)} \cdots \sum_{a_n \in \mathcal{I}(s_n)} \left( \prod_{i=1}^{n} \rho_{f_i}(a_i \mid a_1, \ldots, a_{i-1}) \right) [\![F]\!]_{\nu[x_1 \mapsto a_1, \ldots, x_n \mapsto a_n]}.$$

In the **continuous** case we obtain weighted model integration[25] (WMI) as follows:

$$\int_{a_1 \in \mathcal{I}(s_1)} \cdots \int_{a_n \in \mathcal{I}(s_n)} [\![F]\!]_{\nu[x_1 \mapsto a_1, \ldots, x_n \mapsto a_n]} d\rho_{f_n}(a_n \mid a_1, \ldots, a_{n-1}) \cdots d\rho_{f_1}(a_1)$$

Finally, we can also express even more general dependencies than linear ones. Given any Bayesian network with a set of variables $x_1, \ldots, x_n$, we can express this in ULLER as follows:

$$\sum_{a_1 \in \mathcal{I}(s_1)} \cdots \sum_{a_n \in \mathcal{I}(s_n)} \left( \prod_{i=1}^{n} \rho_{f_i}(a_i \mid \mathrm{parents}(a_i)) \right) [\![F]\!]_{\nu[x_1 \mapsto a_1, \ldots, x_n \mapsto a_n]}.$$

and in ULLER, this is expressed as (assuming that the $x_i$ are topologically ordered, i.e. $x_i$ can only a parent of $x_j$ if $i < j$):

$$x_1 := f_1(), f_2(\mathrm{parents}(x_2)), \ldots, x_n := f_n(\mathrm{parents}(x_n))(F).$$

---

25. Compare with [10].

