# OpenReview forum: "mULLER: A Modular Monad-Based Semantics of the Neurosymbolic ULLER Framework"
_nesyconf.org/NeSy/2025/Conference_Phase_2 — NeSy 2025 - Phase 2 Poster_

### Official Review · Reviewer_3Pt1 · 2025-07-08
**Compelling argument for unifying semantics, but not very accessible**

**Rating:** 6
**Confidence:** 1

**Review:**

Whereas ULLER presented a unified syntax for NeSy systems, Monadic ULLER presents a unified framework for the semantics NeSy systems. It does so by building upon the mathematical concept of monads. The paper contribution is original and significant, but it could gain in clarity.

Strengths:
* **Uniform definition of the semantics through monads**: this seems to make sense and be quite powerful, but as mentioned in the weaknesses, I did not understand all details. I would suggest mentioning that $\text{2CMon-Lat}$, and therefore the derived NeSy systems, have a slightly stronger semantic requirement (commutativity and neutral elements) on the NeSy system than the original ULLER paper ("classical behavior in the limit" when evaluated on discrete 0 and 1 values). These requirements are met by most (all?) existing systems, but this is good to mention for the sake of completeness.
* **The extension to infinite domains**, whose semantics were improperly defined in ULLER, is also a great contribution.

Weaknesses
* **Add more intuitions and concrete examples**: While the paper states to "always begin with a concrete presentation over the category $\textbf{Set}$ that can be understood without any knowledge of category thery" (btw, the notation $\textbf{Set}$ for the category of sets is never introduced), the examples that follow quickly delve into abstract notations without developing intuitions.
A few words around each concept, and a more concrete example with an actual program, its inputs and outputs, would make this a lot clearer. For example:
  * in Example 1, I do not understand if $\eta_X(x) := \delta_x$, which associates to each element $x \in X$ a degenerate distribution with all probability mass in $x$, is supposed to be a simple example of a probabilistic system or if it is defined for all probabilistic systems.
  * I also don't understand the purpose of $f^\ast(\rho)$, can it be understood as computing the expected value of a computation?
  * The categorical generalisation is hard to understand without background in category theory. Possibly, **add a quick background section on Category Theory**.
  * in Defintion 4, what is the strength of a monad? What is $\text{Ob}_\mathcal{C}$?
  * I also found the NeSy shifts hard to understand, perhaps due to missing intuitions early on.

As someone not familiar with category theory, I find the paper quite opaque and unfortunately could not evaluate its soundness. I really wish to understand more of the paper, and I strongly recommend to add more intuitions for the camera ready version especially if the goal is to resonate with as much of the NeSy community as possible.

**Minor questions**:
* In the introduction, the second version of the toy example should be $x:=\textit{dice}()(x=6) \land x:=\textit{dice}()(\textit{even}(x))$ right?
* If you add computational predicates, how do these fit within the scoping concerns of ULLER? For example, $\textit{winLottery}() \land \textit{winLottery}()$ are two independent events according to your semantics? How to have them as one (cf the first dice example in the introduction)? I thought non-determinism is allowed only in statements to be more transparent on this.

**Anonymity:**

Remain anonymous

---

### Official Review · Reviewer_My3J · 2025-07-09
**Monadic ULLER: A Unified Categorical Semantics of the Neurosymbolic ULLER Framework**

**Rating:** 7
**Confidence:** 4

**Review:**

Summary
This paper extends the ULLER language by introducing a monadic categorical semantics. While ULLER was originally designed as a unified language to connect symbolic logic and neural models in neurosymbolic frameworks, its semantics have been somewhat ad hoc, tied to particular logical and neural models. The proposed monadic semantics addresses this limitation by unifying different neurosymbolic systems at the semantic level. In particular, it shows how the previously separate classical, fuzzy, and probabilistic semantics can all be seen as instances of a single categorical framework. This modularity also supports the addition of new semantics and systematic translations between them.

Strengths
* Tackling semantic unification in neurosymbolic systems is an important and timely problem. Building on ULLER, this paper proposes a framework that enables modular addition of new semantics and translations between them, which is a significant step forward.
* The use of monads to unify semantics is elegant and conceptually clean.
* The theoretical arguments appear sound and are well presented.
* The paper is well written and explains relevant categorical, neurosymbolic, and programming language concepts clearly, making it accessible even to readers not deeply familiar with specific concepts.

Weakness
The paper would benefit from implementation examples or at least sketches demonstrating how the modular semantics could be used in practice, e.g., Python library implementation, and how existing common neurosymbolic systems can be unified under the framework.

**Anonymity:**

Remain anonymous

---

### Official Review · Reviewer_1Fh6 · 2025-07-09
**Very strong contents, but possibly too dense for a conference paper**

**Rating:** 6
**Confidence:** 3

**Review:**

The authors consider the ULLER framework for generalising NeSy systems and extend/fix it with a monadic formalism. This allows for defining a single interpretation function over the ULLER syntax, where specific NeSy frameworks are defined only on the level of monads and logical operators (called 'Double Commutative Monoid Lattices', 2CMon-Lat). The authors argue this abstraction will ease implementation.

Strengths:
- The mathematics is quite thorough in specification and abstraction
- This abstraction does allow for more easily defining and comparing different semantics for NeSy systems
- The formalism allows for generalising and fixing some less formal aspects in ULLER. It also reduces the specification of certain semantic rules.
- The individual parts of the paper are relatively clear

Weaknesses:
- The paper is very dense, with many ideas coming at the reader in a short period of time without much contextualisation or examples.
- The paper is not very self-contained: Reading the paper without knowledge of both ULLER and category theory will be very challenging.
- Together, this greatly reduces the accessibility of the paper to the NeSy community.
- There are two methods for using neural networks (in functions and in predicates), which are handled quite differently. If I understand correctly, this results in behaviour that differs from the standard understanding of the frameworks discussed.
- The abstract claims it 'proves' that several semantics are instances of their categorical framework. However proofs of these equivalences are missing.

Ultimately, I believe the contribution of the paper is sufficient for NeSy, and I see no immediate reasons to reject the paper. But I also have multiple questions remaining that should be clarified.

However, my honest opinion is that NeSy 2025 may not be the right venue for this work. The 10 pages greatly limit the authors in explaining their ideas and thoroughly defining the used mathematical concepts. A longer form in a journal would bring across the points much better.

Main questions:
- Definition 6: The main motivation for the original ULLER was reuse of the same syntacticly specified theory + neural networks. However, if I understand correctly, the different semantics in Table 1 require difference interpretations of the computational symbols, since it depends on the choice of $\mathcal{T}$. Doesn't this reduce the reusability?
- As mentioned, the original ULLER did not have computational predicate symbols. Do I understand correctly the regular differentiable fuzzy logic behaviour comes from the computational predicate symbols?
- Relatedly, for probabilistic semantics I would expect a weighted model count when using computational predicate symbols (which output distributions over true/false). However, the probabilities seem to be combined using the product algebra - So the probabilistic semantics actually acts as a fuzzy logic when using computational predicate symbols. The only way to introduce weighted model counting is via the computational function symbols. I don't think this would be intuitive for a standard NeSy person.
- The 'infinitary LTN_p Semantics' 'is a modification of the probabilistic semantics'. This is confusing to me - LTN is a fuzzy logic framework, not a probabilistic one.

Specific comments:
- What is meant with the 'platonic logical world' of objects and truth values?
- Section 2: I did not entirely follow the point about disentangling choices and how Monadic ULLER resolves this (in the end)
- Section 2.1: Please mark this section as background. The text at the beginning suggests the rest of this section is about explaining $x:=m(...$, however this only happens later.
- Example 1: The definition of $\mathcal{D}X$ sums over $X$, however this is only possible if $X$ is countable. I think the sum should be over the support of $\rho$.
- Categorical generalisation: Please specify why this generalisation is needed at this point.
- Example 2: Please specify $\delta_x$ is the dirac measure.
- Definition 3: What is $|mathcal{L}^X$? Remind the reader that $\mathcal{L}$ is a lattice
- Categorical generalisation: Specify what 'internal to' a category means, and what a cartesian closed category is
- Definition 4: Define 'strong monad'
- Definition 5: What does the notation $S^*$ mean? Arbitrary products? It's not related to Kleisli triples, right?
- Definition 5: The families of function symbols  (etc) are confusing. Is the point that there is a single function symbol for each combination of sorts? The subscript does not seem to be used afterwards.
- Definition 7: '$s_T$ is the (unique) sort of the term $T$' how is this derived?
- Definition 7: How are the $\pi$ morphisms defined?
- Section 4.2: Please define the Product BL-Algebra. (Also typo: Prodcut -> Product)
- Section 4.2: Doesn't the product algebra use the Goguen implication instead of the Reichenbach implication? (Eq 4)
- I did not understand the NeSy shifts diagram. Where are these proven? What do the different types of arrows mean? What is id and $\delta$?
- There is a lot of additional material in the appendices, but it is not referenced in the main text. This could help understanding where and when to read it.

Typos:
- Last bullet point in contributions: Use proper quotation marks
- Just before section 2: Thery -> Theory
- Above definition 7: Framworks -> frameworks

**Anonymity:**

Remain anonymous

---

### Official Review · Reviewer_sLXz · 2025-07-09
**Please explain the notation to the non-expert**

**Rating:** 4
**Confidence:** 4

**Review:**

Connecting ULLER with functional programming and potentially with program synthesis would be highly relevant. For this, I find that the paper may be very useful in principle to the NeSy community. However, I believe that presentation in its current state is difficult to access which is a problem if most of the NeSy community won't appreciate the paper. This is particularly problematic with the notation and the main results presented on page 6 and Table 2. Please explain the notation to the non-expert even if that's standard in abstract algebra: if m is a neural net, what is T and what is F?

I'm not sure the main criticisms of ULLER are warranted. What is seen as shortcomings seem to be features, ULLER being intended as a unified library or framework. The claim of "duplication of semantic rules" seems to miss the point of the "library". There are differences in the various NeSy approaches being accounted for. If Category Theory can be a tool to study those differences that's great, but one should not expect to unify everything under the same framework that will then become so abstract as to have no practical value. ULLER is intended as a practical tool. Showing that probability theory and first-order logic all fit within Category Theory doesn't seem to be of great value. When we look at the possible world semantics of Markov models in comparison with many-valued (fuzzy) logic there's a clear difference. I'm not sure there is a lot of benefit in unifying them. Having said that, the discussion of semantics and computation with the proposed conceptualization of argmax is by itself a relevant contribution. Just that point is in my view sufficient for justifying the paper, and it would avoid the more controversial claims that came before. It is good to see the kind of rigorous analysis offered by the paper being derived from the success of the practical tool. I didn't check the correctness of the proof or the appendix, but the overall idea and application of concepts seem valid.

I'd suggest focusing on the above semantics versus computation aspect and providing some examples of the benefit of the framework when applied directly to e.g. LTN and DeepProbLog. This should offer a direct kind of comparative presentation of results that would bring the paper a lot closer to the NeSy audience. I'd love to see more of a discussion of how Haskell could be deployed as claimed in the abstract and more detail provided about the last part of the paper which starts to address the above-mentioned comparative presentation but not with sufficient clarity. Consider for example the way LTN implements the existential quantifier. This is different from Skolemization and it offers in fact a different kind of fuzzy logic-variation that could be studied by the proposed framework.

**Anonymity:**

Remain anonymous